Cellulose aerogel composites as oil sorbents and their regeneration

http://orcid.org/0000-0003-2365-7761 Paulauskiene Tatjana tatjana.paulauskiene@ku.lt
http://orcid.org/0000-0001-8883-0321 Uebe Jochen
Ziogas Mindaugas
Department of Engineering/Faculty of Marine Technology and Natural Sciences, Klaipeda University , Klaipeda , Lithuania
Fernández Marcos Carlos
Electronic publication date: 2021 Aug 3
Publication date: 2021
Volume: 9
Electronic Location ID: e11795
Received 2021 Apr 8; Accepted 2021 Jun 25
Copyright: © 2021 Paulauskiene et al.
Copyright year: 2021
Copyright holder: Paulauskiene et al.
License: This is an open access article distributed under the terms of the Creative Commons Attribution License, which permits unrestricted use, distribution, reproduction and adaptation in any medium and for any purpose provided that it is properly attributed. For attribution, the original author(s), title, publication source (PeerJ) and either DOI or URL of the article must be cited.
License URL: https://creativecommons.org/licenses/by/4.0/

Keywords: Cellulose aerogel composite, Paper and cardboard waste, Sorption capacity, Oil spill, Oil spill clean-up

Funding: The authors received no funding for this work.

==============================
Background

With every oil tanker comes the risk of an accident and oil spill. Sorbents are the most suitable means to remove oil spills. Aerogels as sorbents have high porosity and can be made from cellulose from paper waste. The literature does not distinguish between paper and cardboard as sources of cellulose aerogels and little is known about composites of cellulose aerogels consisting of cellulose fibres and chemically untreated, unprocessed fibres or particles of straw, wool, macroalgae or cellulose acetate from cigarette butts. In this study, the sorption properties for marine diesel oil and biodiesel of such aerogels and their regenerative capacity with bioethanol were investigated.

Methods

Cellulose aerogels were prepared from office paper and cardboard waste without and with chemically untreated algae, straw, wool and cellulose acetate as a composite by freeze drying. All samples were hydrophobised with methylsilane. The density to calculate the porosity and the contact angle were determined. Then the sorption capacity was determined over five cycles of sorption of oil and regeneration with bioethanol.

Results

The average contact angle of all samples was 125°, indicating hydrophobicity. Paper-based aerogels were found to consistently have higher sorption capacities for biodiesel, marine diesel oil and bioethanol than cardboard-based aerogels. In particular, the wool/cellulose aerogel composite was found to have better sorption capacity for biodiesel, marine diesel oil and bioethanol than all other samples. The cellulose acetate/cellulose aerogel composite showed significantly higher sorption capacities than the paper and cardboard control samples (highest value is 32.25 g g−1) only when first used as a sorbent for biodiesel, but with a rapid decrease in the following cycles.

Introduction

As the countries around the Baltic Sea need large quantities of oil and other fuels, there are efforts to transport more cargo with larger ships (Gilek et al., 2016). This increases the risk of accidents and can lead to large amounts of spilled oil (Wang et al., 2014). When a spill occurs, it is important to clean it up as soon as possible to minimise the damage to nature. Oil spills spread under the influence of wind and water currents and undergo a series of chemical and physical changes that are influenced by a number of factors such as oil properties, weather conditions and geography (Saadoun, 2014).

One of the most sustainable methods of removing oil spills is the use of sorbents (Li, Liu & Yang, 2013). As sorbents have been used for many years, there are a number of requirements for their properties: they should be inexpensive, abundant, non-toxic, biodegradable and reusable. In addition, sorbents must have high specificity for the material to be absorbed, high sorption capacity and easy regeneration for reuse (Zamparas et al., 2020; ITOPF, 2012).

Paulauskiene et al. (2014) investigated natural sorbents such as straw, wool, moss, sawdust and peat for oil absorption. These materials achieved a sorption capacity of less than 10 g g−1 for diesel or crude oil. The disadvantage of natural sorbents is that they are dusty, difficult to use in windy conditions and have a low oil absorption capacity. In addition, some natural organic sorbents absorb not only oil but also water, causing the sorbents to sink (Li, Liu & Yang, 2013).

Aerogels as sorbents have a much higher porosity and thus achieve a much higher sorption capacity (Hüsing & Schubert, 1998). In addition, these materials can be subsequently modified by chemical vapour deposition (Chollon, Delettrez & Langlais, 2014) so that, for example, their specificity for non-polar liquids is increased by hydrophobisation (Lin et al., 2015; Li et al., 2014).

Cellulose is a renewable and biodegradable natural polymer most abundant in the world (Long, Weng & Wang, 2018). The solvent of aerogels is replaced by air in the hydrogel, making cellulose-based aerogels environmentally friendly for humans and the environment (Zhang et al., 2015). As a cellulose source for aerogels, there are many possibilities such as plants and plant materials as well as their wastes (Long, Weng & Wang, 2018). The use of paper waste (Nguyen et al., 2013; Paulauskiene et al., 2020), the amounts of which are still increasing (Ferronato & Torreta, 2019; Meng-Chuen Chen et al., 2020), can be considered particularly sustainable, as the paper industry is one of the largest industries in the world (Demirel Bayik & Altin, 2017).

Interestingly, to the authors’ knowledge, cardboard is mentioned as a source of cellulose aerogels in the known literature, but no one has actually explicitly and exclusively used cardboard upon closer inspection. Long, Weng & Wang (2018) mention cardboard once in the introduction. Some primary industrial sources (Zhen et al., 2019; Feng et al., 2015) are probably a mixture of paper and cardboard (Duong & Nguyen, 2016). Paper and cardboard are made from the same raw material, but paper goes through a longer manufacturing process. The raw material for cardboard is less processed, which can also be seen in the coarser fibres when looking at it. Ioelovich (2014) has determined the differences between paper and cardboard in composition. According to this, cardboard contains a significantly higher proportion of lignin of about 18% and hemicellulose of 12%, while office paper contains only about 1% lignin and 5% hemicellulose. The cellulose content is about the same, 61% for board and 62% for office paper. Due to the manufacturing process, paper contains about 30% minerals in contrast to cardboard with about 7%.

Composites are prepared from at least two constituent materials. When aerogel composites are reported, they are predominantly silica aerogel composites (Li et al., 2016; Linhares, Pessoa de Amorim & Duraes, 2019) or so-called polymer aerogels, which are aerogel particles in a polymer matrix (Xiao et al., 2020). This type includes cellulose composites, such as microalgae in a film of cellulose (Yan et al., 2016), chemically treated straw in micropaper (Yousefi et al., 2011) or in polymers (Wang, Qiao & Sun, 2018), enzymatically treated wool in a cellulose acetate matrix (Aluigi et al., 2008) or cellulose-coated wool fibres (Bridgeford & Turbak, 1969; Tran et al., 2016). Surprisingly, to the authors’ knowledge, very little literature is known about composites of cellulose aerogels consisting of cellulose fibres and chemically untreated, virgin fibres or particles of straw, wool, macroalgae or cellulose acetate from cigarette butts. To the authors’ knowledge, only cellulose aerogel composites with graphene oxide have been reported (Zhang et al., 2012).

In this paper, the results of the investigation of aerogels from office paper and cardboard waste as composites with chemically untreated, virgin fibres or particles from straw, wool, algae and cellulose acetate (filter material of cigarettes) were analysed comparatively. Straw, wool, algae and cellulose acetate are easily accessible materials. The aim of their incorporation into cellulose aerogels is to modify the sorption capacities of cellulose aerogels, whereby these materials are incorporated into the cellulose fibres for the aerogels without prior treatment or dissolution. Straw, wool and the macroalga Cladophora were chosen because they are natural and cheap organic sorbents (but often disposed of as waste), biodegradable and their resources are renewable. Cellulose acetate is not natural and biodegrades slowly, but it is cheap. It has already been produced as an aerogel and used as a sorbent for oil (Uebe, Paulauskiene & Boikovych, 2021; Ifelebuegu et al., 2018). The aerogel samples are hydrophobised with a silane reagent, creating a sorbent for oil. Density, porosity and hydrophobicity of the produced aerogels were determined. Furthermore, it was investigated whether an increase in the sorption properties of marine diesel and biodiesel can be measured by the additives. Finally, the regenerative capacity of the aerogels was characterised with ethanol, which is also environmentally friendly and sustainable.

Materials & Methods

Materials used for the experiment

For the production of the aerogels, office paper “Universal” with a grammage of 80 g m−2, produced by The Navigator Company (Portugal), brownish 3-ply corrugated cardboard with a grammage of 603 g m−2, purchased from UAB Dekpaka (Lietuva), and MoTip polyester resin (MOTIP DUPLI Group, Wolvega, The Netherlands) were used as base materials. Trimethoxymethylsilane (MTMS) with a purity of 98% (Sigma-Aldrich Chemie GmbH, St. Louis, MO, USA) was used for hydrophobisation and bioethanol with a purity of 99.5% (Merck KGaA, Gernsheim, Germany) for regeneration. The aerogel additives straw and sheep wool were sourced from a local farm near Klaipeda (Lithuania). Cellulose acetate from cigarette filter production residues was provided by UAB Philip Morris Lietuva. Cladophora macroalgae were collected from the northern beach of Klaipeda (Lithuania) and subsequently air-dried.

For the determination of sorption capacities, marine diesel oil (MDO) with a density of 843 kg m−3 and a dynamic viscosity of 0.0024 Pa s was provided by UAB “Gindana” (Lithuania) and biodiesel with a density of 877 kg m−3 and a dynamic viscosity of 0.0038 Pa s was provided by UAB “Mestilla” (Lithuania).

Preparing the aerogel samples

The scheme of aerogel production and examples of raw materials and additives used for aerogel production are shown in Fig. 1.

Figure 1 Aerogels production scheme; examples of aerogels raw materials and additives: (A) office paper; (B) cardboard; (C) wool; (D) straw; (E) algae; (F) cellulose acetate.

Office paper and cardboard were shredded to a size of 1 cm2, and the cellulose acetate cigarette filters, straw, seaweed and wool were shredded to about 0.5 cm in length. A total of 200 ml of a 1 wt% slurry of the shredded office paper or cardboard in distilled water was then prepared and homogenised for 20 min with a disperser (UltraTurrax T25 digital from IKA, stainless steel rotor/stator diameter of 18 mm). After homogenisation, 1.2 ml (1.1 wt%, ρ = 1.145 g cm−3) MoTip polyester resin and 0.2 g (0.33 wt%) additives such as wool, straw, algae or cellulose acetate were added to the mixture and homogenised for a further 5 min. The mixture was then poured into four 50 mL metal cup and cooled to minus 18 °C for 4 h. The frozen samples were then freeze-dried for 72 h in vacuum at 0.015 hPa and a condenser temperature of minus 105 °C (ScanVac CoolSafe model pro; LaboGene, Bjarkesvej, Denmark). To hydrophobise the samples, MTMS (0.955 g per 1 g aerogel) was added to the bottom of a sufficiently large glass container in a small crystallising dish (Fig. 2) and the samples were arranged vertically above it, separated with a plastic net. The glass container was covered on top with aluminium foil and placed in the oven at 70 °C for 12 h.

Figure 2 Photograph of an aerogel modification: aerogel samples of office paper are white, aerogel samples of cardboard are brownish.

Measurements and calculations of aerogel properties

Determination of the density and porosity

The densities ρ1 of the samples were determined by weighing of a cuboid with known side lengths. Porosity of an aerogel was calculated as follows:

(1) Porosity(1−ρ1ρ2)⋅100,%

ρ2 can be calculated according to the formula as also used by Yan et al. (2016):

(2) ρ2=1Wcelluloseρcellulose+Wadditiveρadditive,gcm−3

wcellulose and wadditive—parts of the cellulose and additive in the sample; ρcellulose and ρadditive it is the densities of cellulose and additives. According to the literature, the density of cellulose is 1.50 g cm−3 (Pereira et al., 2020). The densities of the additives used in the production of the aerogel composites are (ρadditive): algae—1.02 g cm−3 (Zhu, Lu & Dai, 2018), straw—1.1 g cm−3 (Lam et al., 2007), wool—1.3 g cm−3 (Kozyreff et al., 2003), and cellulose acetate—1.3 g cm−3 (dos Santos et al., 2021).

Measurement of the water contact angle

The wettability of the aerogel surface was investigated by measuring the water contact angle θ using the sessile drop method. As described in Paulauskiene et al. (2020), a high-resolution camera of a smartphone was used to take a close-up picture and the contact angle was calculated using the contact angle measurement software (protractor) based on the droplet shape image obtained.

Measurement of the maximum sorption capacity

The maximum sorption capacity of aerogels is a physical property of the aerogel to absorb maximum amount of liquid, in this case oils, into its pores at any given time. First aerogels samples were weighed (m0). Then aerogels samples were immersed in oil (biodiesel or marine diesel oil) for 5 min. The samples were then taken out from the oil, drained for 1 min, and weighed again (m1). The maximum sorption capacity (Q) of the oil was calculated as follows (Li et al., 2018a):

(3) Q=m1−m0m0,gg−1

Ethanol regeneration study on aerogels

Since the maximum sorption capacity for biodiesel and MDO is determined over several cycles, the samples were regenerated with ethanol by the extraction method after each determination.

For the extraction, the samples were placed in a suitable plastic cup with a perforated bottom and washed several times with ethanol. Subsequently, the samples were dried in a drying oven at 70 °C. The samples whose sorption capacity was determined by ethanol were dried in the drying oven immediately after their determination.

Results

Density and porosity calculations of the aerogel samples

Although the same raw materials are used for cardboard and paper, the cellulose aerogels produced from them in this study differ in their properties. The aerogel samples made from office paper and from cardboard without any additives such as straw, wool, algae or cellulose acetate are referred to as control samples in the figures below. The densities of the control samples differ, which is reflected in a slightly different porosity, as shown in Figs. 3 and 4. The densities and porosities of the composites, on the other hand, differ significantly from those of the control samples. They are on average consistently higher and correspondingly the porosities are on average consistently lower than those of the control samples.

Figure 3 Density of aerogels produced from office paper and cardboard with organic additives.

Figure 4 Porosity of aerogels produced from office paper and cardboard with organic additives.

Hydrophobicity of aerogel samples

It is known that paper and cardboard are hydrophilic. Infiltration with MTMS vapour makes the samples hydrophobic. MTMS can react with up to three hydroxy groups of the cellulose and thereby mask the formerly hydrophilic hydroxy groups with a hydrophobic methylsilane group.

The contact angles of all paper and board aerogel samples with and without additives were in the range of 120–125° (Fig. 5). If the angle is less than 90°, the aerogels are hydrophilic, above 90° they are hydrophobic (Ahmad & Kan, 2016).

Figure 5 Water droplet on a sample of aerogel: aerogel photo with water droplet on the top (A), close look to water droplet for contact angle with aerogel (B).

Sorption properties of the aerogel samples

To show that the hydrophobised paper- and board-based cellulosic hydrogel samples with and without additives are suitable for the uptake of hydrophobic liquids, the results of the capacity determinations for biodiesel and MDO are shown below in figure parts a of Figs. 6–9. The reuse efficiencies were calculated for each cycle and shown in figure parts b. The reuse efficiencies of the samples indicate to what extent the previously measured sorption capacity is available again after regeneration with ethanol. For this purpose, the first point of the sorption capacity is taken as a reference point and set to 100%.

Figure 6 Results of the office paper-based aerogels maximum biodiesel sorption capacity (A) and reuse efficiency of the aerogel sample on biodiesel by each cycle (B).

Figure 7 Results of the cardboard-based aerogels maximum biodiesel sorption capacity (A) and reuse efficiency of the aerogel sample on biodiesel by each cycle (B).

Figure 8 Results of the office paper-based aerogels maximum MDO sorption capacity (A) and reuse efficiency of the aerogel sample on MDO by each cycle (B).

Figure 9 Results of the cardboard-based aerogels maximum MDO sorption capacity (A) and reuse efficiency of the aerogel sample on MDO by each cycle (B).

Sorption capacity for biodiesel of the office paper based aerogel samples

The highest measured value of a sorption capacity of all composite samples was obtained by paper-based cellulose aerogels with cellulose acetate in the first application to biodiesel: 32.25 g g−1, the second highest by the composite sample with wool as an additive with 31.98 g g−1 only slightly lower (Fig. 6A). These two samples showed higher sorption capacities than the control sample. In the second sorption cycle, the sorption capacity for the cellulose aerogel/cellulose acetate composite sample dropped below that of the control sample. This trend continued for this sample in the following cycles. This was not the case for the other composite samples, which showed a comparatively gentle decrease in sorption capacity compared to the previous value, while consistently outperforming the control sample.

The reuse efficiency of each sample decreases slowly in this study and seems to approach a constant value for all additives until the fourth cycle. However, in the fifth cycle there is a sudden drop in the case of straw and cellulose acetate as additives and the control sample, most pronounced for straw and the control sample. For cellulose acetate, this drop may not be significant, as the drop in reuse efficiency is strong in all cycles.

Sorption capacity for biodiesel of the cardboard-based aerogel samples

The sorption capacities of the cardboard-based control sample are lower than those of the paper-based sample (e.g. in the first cycle the sorption capacity of the cardboard-based sample is 26.1 g g−1 compared to the paper-based sample with 29.9 g g−1) (Fig. 7). The sorption capacities of these additivated aerogels are generally lower than those of the paper-based samples. With the exception of the first cycle, the values for the sorption capacities of the mixed samples are lower than those of the control sample. Only aerogel samples mixed with cellulose acetate (28.5 g g−1) and this time with straw (26.8 g g−1) show higher sorption capacities than the cardboard-based control sample. This is a clear difference to the paper-based samples. This is an unexpected result, as no distinction is made between paper and cardboard in the literature.

Figure 7B shows the reuse efficiency for algae, straw and cellulose acetate a strong decrease after the first washing cycle, again the strongest is that of cellulose acetate. Compared to the paper-based samples in Fig. 6B, all cardboard-based samples, with the exception of the cellulose acetate-containing sample, are close together in an interval of 59–65% in the fifth cycle, while the paper-based samples are more dispersed in an interval of 49–71%.

Sorption capacity for MDO of the office paper-based aerogel samples

Except for the wool-added cellulose aerogel sample, all other cellulose aerogel composite samples perform worse than the control sample (29.1 g g−1) in the first application (Fig. 8). In the second application, even the algae-added sample shows a higher sorption capacity than the control sample. In the third and fourth application, even straw-added samples show a higher, average sorption capacity. Only the cellulose-acetate-added samples consistently show the lowest sorption capacity in all cycles of this series.

The reuse efficiencies of all samples decrease steadily (Fig. 8B), but in contrast to the previously used biodiesels in Fig. 6B, MDO seems to have a positive effect on the uniformity of the non-linear waste. Even samples additivated with cellulose acetate have a comparatively low drop of only about 20% in the second cycle relative to the initial value. However, the values for the reuse efficiency in the fifth cycle for all samples are significantly lower than the values of the paper-based samples in the biodiesel cycle. The algae-added samples again have the highest reuse efficiency as the paper-based samples in the biodiesel cycle in Fig. 6B.

Sorption capacity for MDO of the cardboard-based aerogel samples

In this test series of the cardboard-based aerogel samples with MDO as test oil, wool as an additive again shows exceptionally high sorption capacities (27.3 g g−1) compared to the pure cardboard-based aerogel (Fig. 9). In contrast to the previous test series, however, by the fourth cycle at the latest all added aerogel samples show a higher sorption capacity than the comparison sample.

The positive influence on the reuse efficiency of the cellulose acetate-based aerogel samples over the sorption-regeneration cycles (Fig. 9B) is also shown by the comparatively high efficiency of 58% in the fifth cycle, which is slightly higher than the paper-based samples in Fig. 8B and almost 15% higher than the cardboard-based samples in the biodiesel cycle in Fig. 7B. Furthermore, again like the previous Figs. 6–8, the sample with algae addition has the highest reuse efficiency in the fifth cycle.

The regeneration of the aerogel samples by ethanol

Since the regeneration of the aerogels takes place between the sorption cycles by washing with ethanol, the following chapter shows that ethanol is absorbed by the hydrophobised aerogels. For this purpose, the sorption capacity over several cycles as well as the reuse efficiency were presented in the same way as for the oils.

Sorption capacity for ethanol of the office paper based aerogel samples

The sorption capacity for ethanol decreases with each additional cycle (Fig. 10A). The capacities are lower than for biodiesel or MDO. But similar to the series of paper-based aerogel samples with biodiesel, wool shows a very high capacity from the beginning (24.4 g g−1), straw and algae surpass the control sample from the second cycle on. Cellulose acetate composite samples show only a comparatively low capacity.

Figure 10 Results of the office paper-based aerogels maximum bioethanol sorption capacity (A) and reuse efficiency (B).

There is a noticeable division into two groups, namely those of the aerogels added with algae, straw and wool and those of the control sample and the sample added with cellulose acetate. In the first group, the values are relatively close to each other during the 5 cycles, while the second group is also relatively close, but drops significantly more (Fig. 6B).

Sorption capacity for ethanol of the cardboard-based aerogel samples

In the first four cycles, the highest sorption capacity was obtained with the cardboard-based aerogel samples containing wool (21.2 g g−1), while the sorption capacities of the algae and straw-doped cellulose aerogel samples were similar to those of the control sample (Fig. 11A). The lowest sorption capacity values in all cycles were obtained with aerogels containing cellulose acetate.

Figure 11 Results of the cardboard-based aerogels maximum bioethanol sorption capacity (A) and reuse efficiency (B).

In Fig. 11B, as with the paper-based aerogels, a division into two groups can also be seen, but the division is slightly different: the group with the better values for reuse efficiency during the cycles consists of the samples with algae and straw additives and the control samples, while the samples with cellulose acetate and wool additives have on average about 10% lower reuse efficiency values. As in all previously presented series with the oils, in Figs. 10 and 11 the algae/cellulose aerogel composite is the sample with the lowest waste and the highest value in the fifth cycle.

Discussion

Density and porosity calculations of the aerogel samples

Although paper and cardboard are made from the same raw material, this raw material is processed differently for these two products. Un-dyed paper is generally much lighter in colour than un-dyed cardboard and consists of thinner fibres. The more intensive refining process changes the composition of the cellulose fibres of the original raw material and results in different properties in the final product (Johansson, 2011; Wathen, 2006; Motamedian, Halilovic & Kulachenk, 2019). This could possibly explain the differences in the densities of aerogels made from pure paper or board (Fig. 3). According to Ioelovich (2014), paperboard consists of 91% polymers such as cellulose (61%), lignin (18%) and hemicellulose (12%) and 7% minerals, while office paper consists of only 68% polymeric components such as cellulose (62%), lignin (1%) and hemicellulose (5%) and 30% minerals.

The porosities of all samples (Fig. 4) are in a range characteristic of cellulose aerogels, as described in Long, Weng & Wang (2018). The paper-based control sample of this study correspond to those in Paulauskiene et al. (2020).

Hydrophobicity of aerogel samples

Cellulose aerogels, when not hydrophobicised, are hydrophilic like their raw material paper and cardboard and allow water to penetrate their highly porous structure. This was shown in Li et al. (2018b), where such aerogel samples serve as sorbents for heavy metal ions, or in Mataar & Boufi (2017), where water-soluble dyes are sorbed. The hydrophobisation of the cellulose samples in this study prevents water penetration, as shown in Fig. 5. The now hydrophobic samples only selectively sorb hydrophobic liquids (Hüsing & Schubert, 1998; Feng et al., 2015), which is also shown by the sorption tests on biodiesel and MDO.

A contact angle between 90° and 150° indicates a hydrophobic interaction between the water droplet and the samples. The measured contact angles indicate the hydrophobicity of the samples and are in the range also mentioned by the literature (Paulauskiene et al., 2020; Zanini et al., 2017) for such samples.

Sorption properties of the aerogel samples

Sorption is the attachment of a liquid to the surface of the fibres and pores of the aerogel. Only van der Waals forces, such as dispersion and dipole interactions, act between absorpt and absorbent. The sorption capacity is determined by the effective surface area and the interstitial space. On the hydrophobised cellulose fibres from the paper and the cardboard as well as the additive, a film of these fluids is formed by the biodiesel, the MDO and also the ethanol. When the effective surfaces are wetted, the interstitial spaces are also filled.

As shown in Fig. 6A, the sorption capacity of the wool and cellulose acetate additivated samples is about 14% higher than the control sample. The control sample shows a comparable sorption capacity as in Paulauskiene et al. (2020). Wool and cellulose acetate thus have a capacity-increasing effect on the sorption of the biodiesel in the first application in Fig. 6A. In the case of the cardboard-based aerogel samples (Fig. 7A), the cellulose acetate also becomes sorption capacity enhancing in the first application, but instead of wool, straw is now the other additive with a slight capacity enhancing effect. In the case of MDO as the sorbent liquid, only wool has a capacity-increasing effect on the first application for both paper- and cardboard-based aerogel samples (Figs. 8A and 9A).

The capacity enhancing effect of the cellulose acetate additive could be explained by the already existing hydrophobicity of cellulose acetate. Cellulose acetate as an independent sorbent with good sorption capacities for oils was demonstrated in Uebe, Paulauskiene & Boikovych (2021). Wool already showed very good sorption properties as a single sorbent material against MDO in Paulauskiene et al. (2014), so that it can be stated here that wool and cellulose aerogels complement each other as a composite. To a lesser extent, however, this also applies to the two other natural materials straw and algae.

The decrease of the sorption capacity in the following applications after the first one and the resulting reduced reuse efficiency (Figs. 6–9B) could be explained by the capillary effect due to the adhesion of oils with the hydrophobic methylsilane groups on the fibres. Under the given conditions, the samples always have an oil film remaining on the effective surface, which the ethanol cannot dissolve out.

In the case of the aerogel composite with straw, algae and wool, there is probably also a reinforcing effect of the cellulose matrix due to the similar morphology of the fibrous additives of the straw, algae or wool. Fibres in composites generally have a reinforcing effect on mechanical stability (Dowling, 2013; Jozwiak-Niedzwiedzka & Fantilli, 2020). This could lead to a reduced effect of the capillary effect during washing and subsequent drying, so that the reuse efficiency increases compared to the control sample.

In the case of cellulose acetate, the poor wettability of this material with water during the manufacturing process may have led to a lack of intimate mixing. The fibrous components of the cellulose acetate particles probably did not bond with the cellulose fibres of the base slurry, but remained as particles in the slurry, which according to Dowling (2013) can lead to rather poor mechanical stabilities depending on the particle size and material. Although cellulose acetate has a sorption capacity-increasing effect during the first application, the sorption capacity is probably brought back to the level of the non-added cellulose matrix or even below during washing and subsequent drying due to the capillary effect.

For a detailed discussion of the differences of the individual samples depending on the composition and the oils used is not possible, because to the author’s knowledge no suitable methods are available for mapping the effective surface and simultaneously distinguishing the cellulose fibres from the additives in the aerogel samples.

A loss of the additives can be excluded, as the oil does not contain any visible residues of cellulose aerogel composite samples. This also applies to the still following cellulose aerogel composite samples. This also means that the cellulose aerogel composite samples can be used freely.

The regeneration of the aerogel samples by ethanol

Comparing the regeneration method of washing with ethanol used in this study with the squeezing method from Paulauskiene et al. (2020), it can be seen that washing is clearly more gentle for the aerogel. Firstly, there is only a comparatively small drop in wool and algae to 70% relative to the initial value and secondly, the drop is continuous. In Paulauskiene et al. (2020), the waste only occurs after the first squeezing to 40% reuse relative to the initial value.

This abrupt drop in the second cycle (Fig. 7B) is similar to the behaviour of the regeneration of cellulose acetate by squeezing in Paulauskiene et al. (2020). The control sample and the one with the additional wool, on the other hand, show a rather slow and steady decrease in reuse efficiency. In contrast to Paulauskiene et al. (2020), the reuse efficiency is higher with the exception of the sample with cellulose acetate, which indicates a much gentler process.

The amount of sorbed ethanol (Figs. 10A, and 11A) is lower in this work compared to marine diesel oil and biodiesel. Cheng et al. (2017) find the same trend and attribute this to the lower density of ethanol compared to MDO and biodiesel. Wang & Liu (2019) shows about equal sorption capacities for ethanol and diesel. Another reason could be the higher polarity of ethanol compared to biodiesel and MDO (Hoffmann et al., 2019) and because of the smaller molecular size, which would lead to smaller adhesion interaction with the methylsilanised cellulose. In the case of the low sorption capacity of cellulose acetate additivated aerogel samples, the insolubility of cellulose acetate in ethanol may also play a role. The decreasing sorption capacity of the aerogel samples for ethanol with each subsequent cycle can be explained by the subcritical drying after use. During the evaporation of the ethanol, the capillary forces between the pore walls and the ethanol are responsible for the shrinking of the sorption capacity (Hüsing & Schubert, 1998).

Conclusions

In summary, cellulose aerogel composites based on paper and cardboard were successfully produced using wool, straw, algae and cellulose acetate and their additive-free control samples. Although the cellulose aerogel literature does not distinguish between paper and cardboard as cellulose sources, significant differences were found between these two types of cellulose aerogels. The sorption capacity of the office paper-based aerogels is higher than that of the cardboard-based aerogels. It can also be stated that the sorption capacity of the cellulose aerogel can be improved by additives. But it probably depends on the processing of the additives with the cellulose slurry. In contrast to fibrous additives, particular cellulose acetate from cigarette butts have low reuse efficiency. It can also be noted that all cellulose aerogel composites can be washed with ethanol to remove absorbed biodiesel and MDO and can thus be regenerated.

In conclusion, cellulose aerogel composites made of cellulose from paper waste mixed with wool or cellulose acetate seem to be best suited to increase the capacity of the sorbent if the sorbent is to be used only once for biodiesel, and only wool if it is also to be used once with MDO. If the sorbent is to be used multiple times, wool or algae would be suitable additives. Wool as an additive for cellulose aerogel composite as a sorbent is most universal for biodiesel or MDO and also for the regenerant ethanol.

Supplemental Information

Supplemental Information 1 Raw Data.

Click here for additional data file.

Additional Information and Declarations

Competing Interests

Author Contributions

Data Availability

The authors declare that they have no competing interests.

Tatjana Paulauskiene conceived and designed the experiments, performed the experiments, analyzed the data, prepared figures and/or tables, authored or reviewed drafts of the paper, and approved the final draft.

Jochen Uebe conceived and designed the experiments, performed the experiments, analyzed the data, prepared figures and/or tables, authored or reviewed drafts of the paper, and approved the final draft.

Mindaugas Ziogas conceived and designed the experiments, performed the experiments, analyzed the data, prepared figures and/or tables, authored or reviewed drafts of the paper, and approved the final draft.

The following information was supplied regarding data availability:

The raw data are available in the Supplemental File.

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
