# Peer review of "Cellulose aerogel composites as oil sorbents and their regeneration"

_PeerJ, doi:10.7717/peerj.11795_

## Round 0.1 · original submission · Minor Revisions

Please, carefully address all the changes suggested by the reviewers. Revise the English all through the paper. Also revise the figures’ format and captions, as outlined by reviewer 1. As the behaviour of ethanol would be the same, regardless of its origin, it is unnecessary to repetitively use the term “bioethanol”. This can be clarified in the Materials section and maybe in one of the headings and use just “ethanol” in all the other entries. The actual amounts of each component in the preparation of the aerogels should be specified together with its percentage in the Materials and Methods section. Figure 1 is confusing, as it gives some of the components as percentage (i.e., 1% paper in water) and some as absolute amount (i.e., 6 ml polyester, 0.1 g additive). At least, the volume of water used to prepare the initial paper slurry should be specified. Also, the authors should provide a justification of why that particular proportion of additives has been used and discuss if a better performance of the composites would be expected with different amounts of additives.

Reviewer 1 ·

Basic reporting

The article introduces interesting findings about cellulose aerogel composites as oil sorbents. These are not very breakthough ones but meaningful for industry and can be good reference for other studies. Literature references and background are sufficiently provided.

Experimental design

The research question is well defined and the method described with sufficient information. But more details should be provied if possible (see the general comments)

Validity of the findings

Please review the Figures 6 to 17. Each figure only has part (a) (or part (b)). And the caption of some figures are the same, such as figure 16 and figure 17.

Additional comments

- Can the author explain the measurement of the water contact angle in more details. For example, which device was used to measured the angle? How did it calibrated?
- In the graphs (Fig.6 to Fig.17),the legend can be put into white backrgound text box to avoid being cut through by the grid lines
- Line 206, please check the range of contact angles 120-125 degree. In the next line, the authors mention that the highest is 137.8 degree?
-

·

Basic reporting

English should be improved for the clearly understanding of the paper.

Experimental design

No comment

Validity of the findings

No comment

Additional comments

The manuscript described the study on preparation of cellulose aerogel composites for oil sorbents application. The data is interesting and is suitable to the journal. However, it need further experiments and data so that the content of the research is clarified and the explanation of obtained result is more proper.

1. Specific surface area measurement should be carried out. It helps to explain the adsorption of the materials more clearly than the porosity solely does.
2. Stability of aerogel when oil was absorbed in the sample should be investigated. For example: evaluate the mechanical properties of sample after each time of absorption.
3. SEM image should be provided to clearly observe the structure as well as the surface of sample. This may be helpful in the explanation of contact angle because contact angle was predominantly dependent on the surface structure.

---

## Round 0.2 · accepted · Accept

As you can see, both reviewers are satisfied with the way you handled their suggestions. Your article is now ready to be published.

Reviewer 1 ·

Basic reporting

the content of this article is clear and well-written.

Experimental design

method has been sufficiently updated

Validity of the findings

no comment

Additional comments

The authors provide acceptable answers for the questions and comments. The paper is recommended to be published on this journal.

·

Basic reporting

No comment

Experimental design

No comment

Validity of the findings

No comment

Additional comments

The authors revised the manuscript according to the reviewer's comments. The manuscript is acceptable.